# Recruitment of Scc2/4 to double-strand breaks depends on γH2A and DNA end resection

Martin Scherzer, Fosco Giordano, Maria Solé Ferran, Lena Ström ⓘ

Homologous recombination enables cells to overcome the threat of DNA double-strand breaks (DSBs), allowing for repair without the loss of genetic information. Central to the homologous recombination repair process is the de novo loading of cohesin around a DSB by its loader complex Scc2/4. Although cohesin's DSB accumulation has been explored in numerous studies, the prerequisites for Scc2/4 recruitment during the repair process are still elusive. To address this question, we combine chromatin immunoprecipitation-qPCR with a site-specific DSB in vivo, in *Saccharomyces cerevisiae*. We find that Scc2 DSB recruitment relies on γH2A and Tel1, but as opposed to cohesin, not on Mec1. We further show that the binding of Scc2, which emanates from the break site, depends on and coincides with DNA end resection. Absence of chromatin remodeling at the DSB affects Scc2 binding and DNA end resection to a comparable degree, further indicating the latter to be a major driver for Scc2 recruitment. Our results shed light on the intricate DSB repair cascade leading to the recruitment of Scc2/4 and subsequent loading of cohesin.

## Introduction

A cornerstone in the maintenance of genomic integrity is a cell's ability to repair DNA damage. This encompasses an orchestrated series of events necessary to keep its genetic material intact. DNA double-strand breaks (DSBs) pose the most hazardous threat to genomic integrity, rendering repair of the same of vital importance (1). The repair of DSBs is mediated by two major pathways, comprising non-homologous end joining (NHEJ) and homologous recombination (HR). Whereas NHEJ offers rapid repair based on direct end joining, it correlates with increased risk for erroneous repair. HR on the other hand depends on a suitable repair template and is primarily restricted to the S-G2/M-phases of the cell cycle (2, 3).

The HR repair pathway relies on proteins of the highly conserved *RAD52* epistasis group (4). Among them are Mre11, Rad50, and Xrs2, the constituents of the MRX complex. Credited with the recognition of DSBs, MRX is recruited to broken DNA ends, initiating early stages of DNA end resection and providing a binding platform for the effector kinase Tel1. Recruitment of Tel1 facilitates checkpoint activation, phosphorylation of histone H2A and prevents further progression through the cell cycle (5). The absence of NHEJ-directing Ku proteins in postreplicative cells then allows the initiation of long-range DNA end resection, carried out by the exonucleases Dna2 and Exo1 (6). The resulting 3′ overhang ssDNA ends are rapidly bound by replication protein A (RPA) and provide a binding platform for Ddc2, enabling the recruitment of the Mec1 kinase (7), which in turn reinforces checkpoint activation (8). The following replacement of RPA with Rad51 enables search for a repair template and the subsequent synthesis of the lost sequence (9). The presence of a sister chromatid in S-G2/M provides cells with a bona fide template to amend DSBs, with minimized loss of genetic information.

Central to the organization of sister chromatids is the ring-shaped protein complex cohesin. The core cohesin complex consists of the Smc1/Smc3 heterodimer ATPase, as well the kleisin subunit Scc1 and the HEAT repeat protein Scc3. Capable of entrapping DNA within its ring, cohesin is loaded onto DNA in the early S-phase by the separate loader complex composed of the proteins Scc2 and Scc4 (Scc2/4). Cohesin chromatin association is also influenced by its HEAT repeat containing regulatory factors Wpl1 and Pds5 (10, 11). Upon acetylation of the Smc3 subunit by the acetyltransferase Eco1, cohesin binding is stabilized, admitting tethering of sister chromatids in a process referred to as establishment of cohesion (12). Once loaded, ATP hydrolysis allows cohesin to relocate from the sites of loading and Scc2/4 itself (13). This was demonstrated by calibrated chromatin immunoprecipitation (ChIP)-Seq, where only a weak correlation between cohesin and Scc2/4 binding was found, unless cohesin's ATPase function was impaired (14). In anaphase, after formation of the mitotic spindle, the Scc1 subunit is cleaved by separase, allowing the segregation of sister chromatids (15). As opposed to yeast, where cohesin keeps the chromosomes cohesive along their entire lengths until anaphase, in higher eukaryotes, most of the chromosome arm–bound cohesin is removed by separase-independent means already in prophase, leaving only centrometric cohesin subjected to cleavage of Scc1 (16).

The cohesin complex was first recognized for its significance in DNA DSB repair (17). It was later shown that cells were unable to

---

Department of Cell and Molecular Biology, Karolinska Institutet, Stockholm, Sweden

Correspondence: lena.strom@ki.se
Maria Solé Ferran's present address is Spanish National Cancer Research centre (CNIO), Madrid, Spain.

repair DNA damage if they failed to establish cohesion during S-phase (18). Subsequent experiments demonstrated that post-replicative de novo binding of cohesin occurs around the break, spanning a region of about 50 kb (19, 20). In line with its loading in the early S-phase for sister chromatid cohesion, this process relies on Scc2/4. It was also shown that NIPBL and MAU2, the human homologs of Scc2 and Scc4, respectively, localize to laser microirradiation–induced DNA damage stripes as well as Fok1-generated DSBs (21, 22); however, DNA binding at breaks has only formally been demonstrated for Scc2 in yeast (23). Mounting evidence (24, 25, 26) attributes the cohesin loader with a central role in the DNA damage response, yet prerequisites for its recruitment to DSBs are still elusive, despite numerous factors influencing accumulation of cohesin at DSBs having been identified (19, 20).

Here we address the requirements for Scc2 recruitment to DNA DSBs in yeast using chromatin immunoprecipitation to assess its binding, in selected genetic mutants. We find that the recruitment is largely driven by Tel1 and phosphorylation of histone H2A, whereas Mec1, contrary to its significance for cohesin loading, is dispensable for this process. We demonstrate that cohesin loading by Scc2/4 occurs directly at the break, suggesting that translocation of cohesin determines its observed binding profile. Furthermore, we show that Scc2 binding coincides with DNA end resection, where delayed or accelerated end resection affects Scc2 recruitment in a corresponding manner. The significance of chromatin remodelers for this recruitment appears to be limited to their impact on resection. We conclude that DNA end resection together with γH2A are driving factors for Scc2/4 recruitment to DSBs, yet by themselves insufficient to facilitate cohesin loading. These findings provide insight into the sequence of events essential for Scc2 recruitment and cohesin loading during the DSB repair and suggest a potential cohesin-independent role for Scc2/4 at DSBs.

## Results

### Spatial distribution of Scc2 in the vicinity of a site-specific DSB

Cohesin's dependence on Scc2/4 as a loading factor has been well established, both in vivo (27) and in vitro (28). Likewise, it was shown that de novo loading of cohesin at DSBs in postreplicative cells relies on the presence of Scc2 (19, 20). In agreement with this and fortified by previous Chip-on-chip data in budding yeast, Scc2 can be readily detected at elevated levels around a DSB after 90 min of break induction (23), although the requirements for its recruitment are still poorly investigated. We have previously shown that NIPBL, the human homolog of Scc2, localizes to Fok1 generated DSBs through interaction with HP1 (21). However, the absence of an HP1 homolog in *Saccharomyces cerevisiae* (29) raises the question how this recruitment occurs in yeast. To gain insight into this matter, and further extend the general understanding of Scc2 DSB recruitment, we combined an inducible Homothallic (HO)-endonuclease system with ChIP-qPCR to assess the binding of Scc2/4 around a DSB. An ectopic HO recognition sequence was introduced on Chromosome VI (206,1 kb from the left telomere) in equal distance from the centromere and the right telomere, respectively. To make our data comparable between

experiments, we used previously published low-binding sites of Scc2 on Chromosome V for normalization of the qPCR results (30). The general experimental layout is illustrated in Fig 1A.

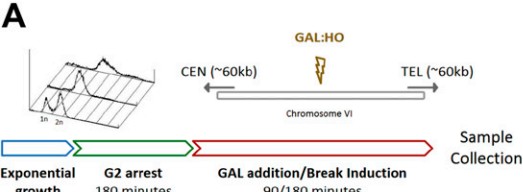

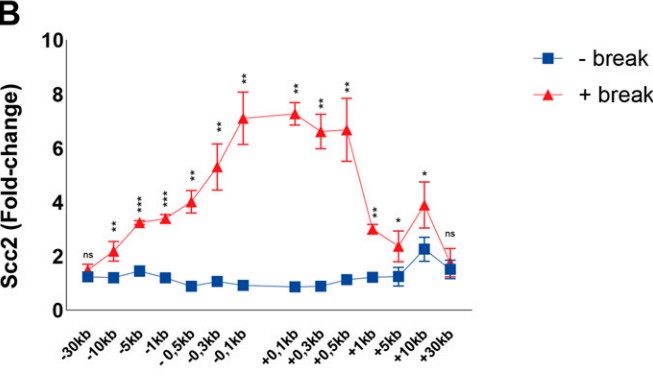

**Figure 1. Scc2 is recruited to DNA double-strand breaks with a binding pattern distinct from Scc1.**
**(A)** Schematic of the basic experimental setup used for all chromatin immunoprecipitation (ChIP)-qPCRs, unless otherwise stated. Cells in exponential growth phase were arrested in G2/M by addition of benomyl for 3 h. Galactose was added to induce a DNA DSB on the arm of chromosome VI in equal distance to the centromere and telomere, or not. At indicated time points samples were collected and binding of the protein of interest was assessed by (ChIP)-qPCR. Primer pairs used are indicated in the following figures according to their distances upstream (−) and downstream (+) from the DSB site. Cells were grown at 25°C throughout. **(B)** Scc2 binding is significantly elevated up to 10 kb around a DSB, 90 min after break induction (red), compared with no break (blue). Data were normalized to known low-binding sites for Scc2 (93%). **(C)** Scc1 binding is significantly elevated at all observed loci except at −0.3 kb, 90 min after break induction (red), compared with no break (blue) (95%). **(B, C)** The graphs show means and SD of (B) n = 3 and (C) n = 2. *t* test was used to compare +break and −break at respective locations. Significance: *P < 0.05; **P < 0.01; ***P < 0.001; ns, not significant. Data were adjusted to the average cut efficiency for respective strain shown in squared brackets.

Logarithmically growing cells were arrested in G2/M, when galactose was either added to the culture (+break) to generate a DSB, or not (–break), for comparison with Scc2/4 binding in unchallenged conditions. Samples for ChIP qPCR were collected 90 min later. At this time point, binding of Scc2/4 was significantly increased up to 10 kb around the DSB, compared with unchallenged conditions (Figs 1B and S1A). This accumulation was comparable on both sides and increasing towards the break site. Interestingly, the most prominent binding of Scc2 and Scc4 occurred within 1 kb of the DSB. Because Scc2 and Scc4 displayed highly similar binding patterns in response to the DSB, in agreement with studies analyzing their chromatin association in unchallenged cells (13), we will here focus on Scc2.

The abundant binding of Scc2 within 1 kb of the DSB was surprising because it is contrasting the reported binding pattern for cohesin (19, 20). We therefore determined binding of Scc1 in the presence and absence of a DSB after 90 min with our experimental system. This showed overall elevated levels of Scc1 up to 30 kb on both sides of the DSB, with the highest absolute increase found at previously identified cohesin binding sites, present also under unchallenged conditions, at –5 and +10 kb from the break (Fig 1C). As opposed to Scc2, cohesin's association did not increase excessively adjacent to the DSB (compare Fig 1B and Fig 1C).

Previous investigations have shown a competing relationship between Pds5 and Scc2 for cohesin loading (31). To address if this opposing relationship can be seen during DNA damage, we analyzed Pds5's chromatin association at a DSB. In agreement with previous reports, Pds5 colocalized with Scc1 under unchallenged conditions at –5 and +10 kb from the cut site, but contrary to cohesin or Scc2 failed to be significantly elevated in response to the DSB (Fig S1B). This might indicate that Pds5 preferentially associates with S-phase loaded cohesin, rather than de novo DSB loaded cohesin. However, although not differentially regulated in response to a DSB, a previous study has shown the absence of Pds5 to affect γH2A spreading in response to a break (32).

Although cohesin depends on Scc2/4 as a loading factor, ChIP-Seq profiles of these complexes exhibit only minor colocalization in vivo. This has been credited to cohesin's ability to translocate along DNA, where ATP hydrolysis qualifies cohesin to vacate its initial loading sites (14). To address this phenomenon in the context of a DSB, we used a previously described transition state mutant allele of SMC3 (smc3-E1155Q), capable of binding DNA and ATP, but unable to hydrolyze the latter, rendering cohesin "immobilized" at its sites of loading (14). Induction of a DSB for 90 min leads to accumulation of wild-type Smc3 around the break up to 30 kb (Fig S1C), with the most obvious increase in binding at –5 and +10 kb from the break, comparable with previous observations of Scc1 (Fig 1C). However, a noted increase adjacent to the break was also observed. In contrast, the accumulation of Smc3 harboring the E1155Q mutation was confined to the immediate proximity of the DSB up to 1 kb (Fig S1D), comparable with our observations made for Scc2. This together suggests that de novo loading of cohesin occurs at the DSB ends, although we cannot fully exclude intermittent loading sites between our investigated loci. ATP hydrolysis then enables cohesins translocation towards more distal regions, whereas the most substantial increase in Scc2 recruitment remains in the vicinity of the DSB. Nevertheless, because the resolution of ChIP is limited by

shearing efficiency (~300–700 bp), we decided to investigate the requirements for Scc2 binding from 1 kb and outwards on the left side of the DSB. The accumulation of Scc2 in a non-mutant strain will henceforth be referred to as wild type.

## DSB recruitment of Scc2 relies on Tel1 but not Mec1

DSBs are rapidly recognized by the DNA damage sensing Mre11-Rad50-Xrs2 (MRX) complex (33). Components of the MRX complex have previously been shown to affect DSB recruitment of yeast and human cohesin (20, 34, 35). To investigate the requirement of MRX for Scc2 recruitment, we assessed Scc2's binding in strains lacking either Mre11, Rad50, or Xrs2. Our data demonstrate that binding of Scc2 within 5 kb of the DSB was significantly reduced in mre11Δ and rad50Δ cells, whereas recruitment in xrs2Δ was diminished up to 10 kb (Fig 2A), rendering the MRX complex an integral part in the recruitment of Scc2. This recruitment did not depend on the nuclease activity of Mre11 as binding was unaffected by two different Mre11 nuclease deficient alleles (Fig S2A).

The initial response to a DSB is accompanied by activation of the DNA damage checkpoint, a process largely regulated by the two kinases Tel1 and Mec1 (36). During checkpoint activation, Tel1 is recruited first, showing a high affinity for broken blunt DNA ends (37), whereas recruitment of Mec1 occurs later (5, 38). Tel1 was previously reported to be recruited to the DSB by the C-terminus of Xrs2 (39). In agreement with this, deletion of Tel1 led to a considerable reduction in Scc2 recruitment to the break site, comparable with the absence of Xrs2 (Fig 2B).

Among the first targets of Tel1 is phosphorylation of histone H2A at Ser129, referred to as γH2A from here on (40). γH2A has been recognized as an early signal of DNA damage in eukaryotes and is required for the assembly of subsequent effector molecules (41). In line with previous observations for cohesin (20), recruitment of Scc2 was indeed impaired (Fig 2C) in strains harboring non-phosphorylatable mutations in both homologs of the H2A gene (hta1-S129A and hta2-S129A) (42).

We next asked, whether absence of Mec1, the second master kinase, augmenting phosphorylation of H2A, would yield similar results as seen for Tel1 (Fig 2B). To our surprise, deletion of MEC1 had no discernible effect on Scc2 recruitment to the DSB (Fig 2D). These results were quite unexpected, as de novo loading of cohesin at the DSB was previously shown to rely on the presence of Mec1, more so than Tel1 (20, 43).

To validate this apparent and potentially interesting discrepancy between Scc2 and cohesin, we wanted to confirm the importance of Mec1 for cohesin recruitment. In agreement with published results (20, 38), deletion of Mec1 resulted in a significant reduction of cohesin recruitment to the DSB, compared with wild type (Fig 3A, compare Fig S3). These data suggest different prerequisites for recruitment of Scc2/4 and the subsequent loading of cohesin.

Because Mec1 recruitment to DSBs relies on RPA-bound single-stranded DNA (8) generated by DNA end resection, this opened for the possibility that Scc2 in itself could be important for the resection process. To test this hypothesis, we generated a strain carrying an auxin-inducible degron (AID [44]) allele of Scc2, under the control of a Tetracycline regulated promoter (45). Addition of doxycycline and auxin after the cells were arrested in G2/M led to a

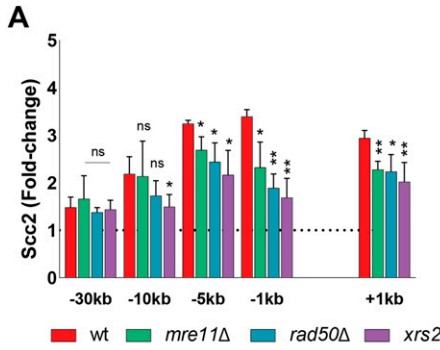

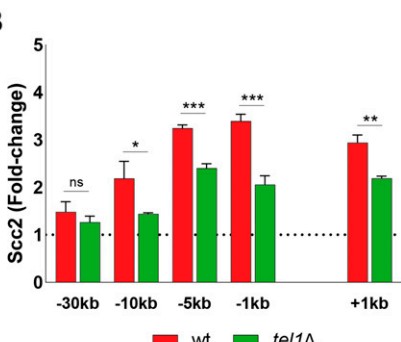

**Figure 2. Double-strand break recruitment of Scc2 depends on the MRX complex, γH2A, and Tel1 but not Mec1.**
**(A, B, C, D)** Chromatin immunoprecipitation-qPCR of Scc2 (93%) binding at the DSB in (A, B, C, D) wild type and strains lacking (A) Mre11 (84%), Rad50 (90%), or Xrs2 (85%), (B) Tel1 (95%) (C) non-phosphorylatable alleles of histone H2A (95%) or (D) Mec1 (94%). The graphs show means and SD of n = 3. *t* test was used to compare normalized values of Scc2 between wild type and indicated mutants at respective locations, 90 min after break induction. Significance: *P < 0.05; **P < 0.01; ***P < 0.001; ns, not significant. Data were adjusted to the average cut efficiency for respective strain shown in squared brackets.

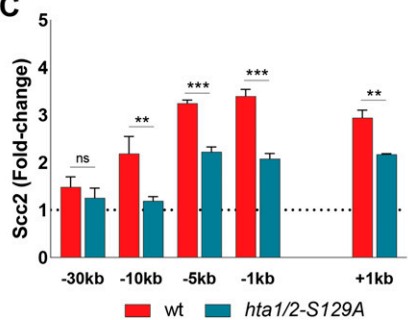

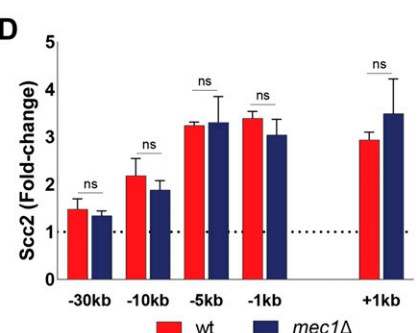

substantial reduction in Scc2 protein levels, which remained low during the course of the experiment (Fig 3B, right). To confirm the lack of Scc2 functionally, we analyzed the accumulation of cohesin in the presence and absence of a DSB with and without Scc2. Absence of Scc2 indeed resulted in lack of de novo cohesin loading in the presence of a DSB, compared with control conditions (Fig 3B, left). This resembles what has previously been reported to be a consequence of inactivating the Scc2 temperature sensitive allele, *scc2-4* (19, 46). Having the possibility to degrade Scc2 efficiently, we next investigated whether Scc2 influenced the degree of resection at the DSB, to exclude an indirect effect on Mec1. For this, we adapted a qPCR-based assay from Zierhut and Diffley (47). Resection profiles in the presence of Scc2 were comparable with previous studies, reaching around 90% of resected DNA after 6 h of break induction 8 kb from the DSB (48). Absence of Scc2 did, however, not affect the rate of DNA end resection (Fig 3C), making an indirect effect on Mec1 recruitment unlikely.

### DNA end resection is a driving factor for Scc2 recruitment

Considering the modest effect of Mec1 on Scc2 recruitment, and Scc2 being insignificant for resection, we decided to investigate the importance of resection for the recruitment of Scc2 to DSBs. To monitor the recruitment of Scc2 in the context of DNA end resection we followed the accumulation of Scc2 over a 6-h period after break induction, assessing its binding in 90-min intervals. Ongoing break induction led to a constant increase in Scc2 around the break and elevated levels of Scc2 at −30 kb from the break site after 6 h (Fig 4A). This increase over time was more prevalent closer to the break. Because of the limitation of the qPCR-based approach to measure

ssDNA, relying on restriction enzyme cut sites and enzyme efficiency, we instead decided to assess DNA end resection by using RPA ChIP as a readout. Our data show that the RPA binding pattern resembled that for Scc2 (Fig 4B), suggesting that recruitment of Scc2 coincides with and might depend on DNA end resection, as has been shown for its binding at stalled replication forks (49). This prompted us to increase the time of break induction in the following experiments, to allow for DNA end resection. Given the speed of end resection of around 4–5 kb/h (50, 51) and the kinetics of break formation, we decided to analyze the recruitment of Scc2 after 180 min from here on, as resection proceeds well beyond 10 kb (Fig 4B) in most cells during this time. To assess the impact of DNA end resection on Scc2 recruitment, we analyzed its binding in a strain carrying deletions of *SGS1* and *EXO1*, noted for a severe resection deficiency beyond a few hundred base pairs. Recruitment of Scc2 to the DSB was significantly decreased in the *sgs1Δexo1Δ* background (Fig 4C).

If resection as such is a determining factor for Scc2 recruitment, then increased resection should augment the Scc2 binding. To test this, we assessed Scc2 recruitment in a strain lacking Rad9 which causes DNA end resection to proceed at an accelerated pace (52). Consistent with our hypothesis, deletion of *RAD9* resulted in significantly elevated levels of Scc2 recruitment to the DSB compared with wild type (Fig 4D). Based on these results, we conclude that DNA end resection is a decisive process for Scc2 recruitment, whereas loading of cohesin requires additional events to take place, such as presence of Mec1 (Fig 3A) (20).

A recent study by Arnould et al (32), investigating the role of topologically associating domains in DNA damage repair, proposed a model where a loop extruding mechanism allows rapid

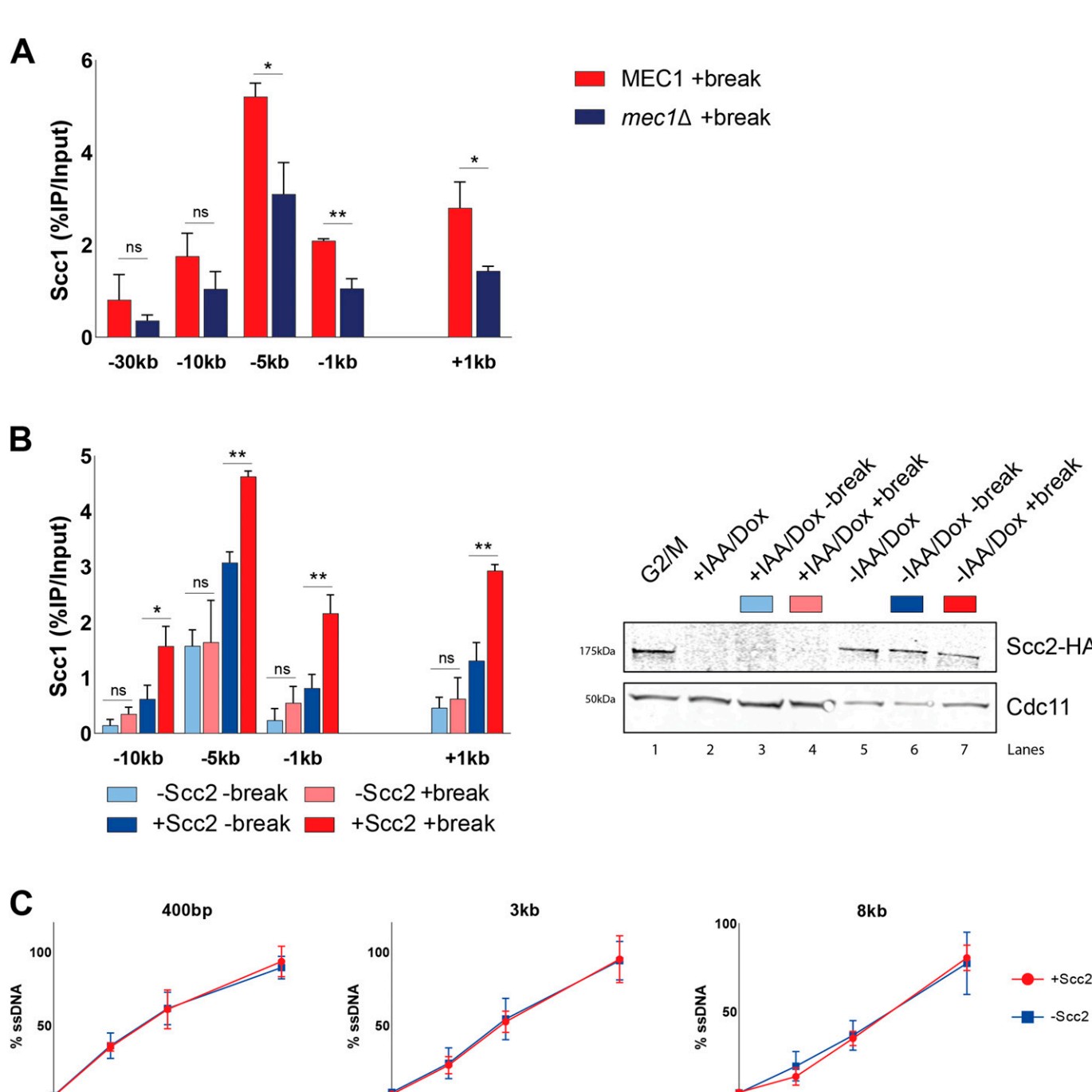

**Figure 3. Cohesin loading by Scc2/4 at double-strand breaks relies on Mec1.**
**(A)** Chromatin immunoprecipitation (ChIP)-qPCR of Scc1 DSB binding in a wild type and a *mec1Δ* strain (>99% and 96%). **(B)** Left: (ChIP)-qPCR of Scc1 binding at the DSB in the presence or absence of Scc2 in an Scc2 degron strain (99% and 99%). Cells were grown and arrested as indicated in Fig 1. Before break induction, cultures were split in two, with one half receiving auxin and doxycycline and the other half a corresponding amount of EtOH for 2 h to degrade Scc2 or not. Each culture was then split again, totaling four and receiving galactose or not. Right: Western blot showing protein levels of Scc2. Protein samples were taken after 3 h arrest (G2/M, lane 1), subsequent 2 h of either IAA/Doxy (+IAA/Dox, lane 2), or EtOH (−IAA/Dox, lane 5) incubation, and following 3 h of either break induction in the presence (−IAA/Dox +break, lane 7) or absence of Scc2 (+IAA/Dox +break, lane 4), or under no break condition in the presence (−IAA/Dox −break, lane 6) or absence of Scc2 (+IAA/Dox −break, lane 3). Cdc11 served as a loading control. **(C)** Measurement of ssDNA at the DSB in the presence (red) (93% at 90′ and >99% onward) and absence (blue) of Scc2 (93% at 90′, 99% at 180′, and >99% at 360′) at indicated distances from the break. **(B)** Cells were grown as in (B), and samples collected after addition of galactose for 0, 90, 180, and 360 min. **(A, B, C)** The graphs show means and SD of (A) n = 3, (B) n = 2, and (C) n = 2. **(A, B)** *t* test was used to compare values of Scc1 between (A) wild type and *mec1Δ* or (B) +break and −break in the presence or absence of Scc2 at respective locations, 180 min after break induction. **(C)** No significant difference was observed at any

phosphorylation of histone H2AX as DNA is reeled in by DSB an-chored cohesin. Our observation that Scc2 "emanates" from the break site could potentially be explained by cohesin-mediated similar mechanism (Fig S1D). To address the possibility that Scc2/4 would load cohesin at the break site and then be shuttled away by cohesin, we used a strain carrying an Scc1-AID construct. This allowed for degradation of Scc1, which consequently would interfere with potential cohesin-mediated loop extrusion at the break. However, degradation of Scc1 before break induction (Fig 4E, right) caused no apparent reduction in the recruitment of Scc2 to the DSB (Fig 4E, left). Based on these results, we conclude that both the recruitment and the "emanation" of Scc2 at DSBs occur inde-pendently of cohesin.

## Impact of chromatin remodeling on Scc2 recruitment

To validate the importance of end resection for Scc2 recruitment, we next decided to address the role of chromatin remodelers, as DNA end resection is tightly regulated by chromatin remodeling (53). Alternatively, chromatin remodelers could also be directly responsible for the recruitment of Scc2. To investigate this we channeled our attention first to the RSC complex. It was previously demonstrated that the Sth1 ATPase subunit of the RSC complex acts as a chromatin receptor, facilitating the binding of Scc2/4 and subsequent loading of cohesin (54). Given the significance of the RSC complex for loading of Scc2/4 during an unchallenged cell cycle and its central role in the early processing of DSBs (55), we asked whether Sth1 was equally integral for the recruitment of Scc2 to DSBs. To test this, we used a strain expressing AID-tagged Sth1 under a repressible *MET3* promoter (54). Presence of methionine and auxin reduced the Sth1 protein level by more than 80%, before break induction (Fig S4A). Cells grown under permissive conditions failed to arrest in G2/M due to poor growth in minimal media without methionine, combined with raffinose as a suboptimal carbon source. We therefore assessed Scc2 levels in a wild-type strain exposed to the same experimental conditions as the strain harboring the degradable Sth1 allele. Whereas the overall re-cruitment of Scc2 was reduced compared with wild type (Fig 5A, left), binding increased in response to a DSB, despite the absence of Sth1, allowing for the argument that the role of the RSC complex for Scc2/4 loading does not extend to the DNA damage repair response. We reasoned that the reduction in recruitment is rather due to in-direct effects on DNA end resection (55). To validate this, we assessed degree of DNA end resection by RPA coverage. RPA binding was significantly reduced up to 5 kb from the DSB in the absence of Sth1 (Fig 5A, right), fortifying the possibility that the reduction of Scc2 recruitment to the DSB may be a consequence of impaired DNA end resection.

Next, we asked if other chromatin remodelers could be re-sponsible for the recruitment of Scc2 to DSBs. Considering the significance of γH2A for the recruitment of Scc2 (Fig 2C), the SWR1-C and INO80 chromatin remodeling complexes posed as suitable candidates because both have been shown to be able to bind to γH2A (56, 57, 58). Absence of Swr1, the ATPase subunit of SWR1-C, lead to a significant reduction in Scc2 at the DSB (Fig 5B, left). Although described as resection-proficient (59), RPA coverage in swr1Δ confirmed perturbed generation of ssDNA under these ex-perimental conditions (Fig 5B, right). Although cut efficiency at the experimental end point was appreciable (Fig S4B), one reason for these conflicting results could be explained by delayed HO en-donuclease kinetics. In addition to being responsible for the in-corporation of the histone variant H2A.Z (Htz1) at DSBs (60), deletions of *SWR1* and *HTZ1* have further been linked to genome-wide transcriptional misregulation (61). This is supported by failed efforts to achieve observable break induction in an htz1Δ back-ground within 3 h (Fig S4C).

Because the ATPase subunit of INO80 is essential in W303 (62), we decided to address its role for recruitment of Scc2 using a strain harboring a deletion of the Nhp10 subunit, which interferes with the recruitment of INO80 to DSBs (63). Noted for deficient DNA end resection (59, 64), deletion of *NHP10* resulted in a significant re-duction in the recruitment of Scc2 to the DSB at −5 and −1 kb (Fig 5C, left). In agreement with the observed binding pattern for Scc2, nhp10Δ resulted in a reduction of RPA filament formation around the break (Fig 5C, right). These results demonstrate that recruitment of Scc2 to DSBs is impaired by deficient chromatin remodeling at the break. However, as the reduction of Scc2 matches the degree of resection, we reason, although not formally excluding, that Scc2 loading is unlikely to be directly facilitated by the remodeling complexes probed here.

## RPA and Rad51 impede binding of Scc2 at the DSB

We next decided to address the role of RPA itself, as it has served as a respectable indicator of Scc2 recruitment. Integral to the pres-ervation of ssDNA at broken DNA ends, it has been shown that RPA enforces the correct polarity of DNA end resection and promotes the recruitment and activity of Dna2 (65). To this end, we made use of the *rfa1-G77E* allele, characterized as the *Saccharomyces cere-visiae* equivalent to the fission yeast *rfa1-G78E* allele (66, 67), shown to exhibit markedly reduced affinity of RPA for short ssDNA frag-ments, and a slightly reduced affinity for longer stretches. Inter-estingly, Scc2 recruitment was significantly increased close to the DSB ends, whereas recruitment was unaffected at more distal regions (Fig 6A). This increase of Scc2 accumulation could suggest a competing relationship between RPA and Scc2 for ssDNA, similar to what has been demonstrated for RPA and cohesin (66). Previous studies have demonstrated that the absence of RPA results in failure to form Rad51 filaments (68). This prompted us to investigate what impact Rad51 has on the recruitment of Scc2. In agreement with this, the absence of Rad51 similarly caused an increase in the recruitment of Scc2 proximal to the break, comparable with the *rfa1-G77E* mutant (Fig 6B). These results allow for several conclu-sions. Although RPA coverage was indicative for Scc2 binding in

---

distance at any given time point using *t* test. Significance: *P < 0.05; **P < 0.01; ***P < 0.001; ns, not significant. Data were adjusted to the average cut efficiency for respective strain shown in squared brackets.
Source data are available online for this figure.

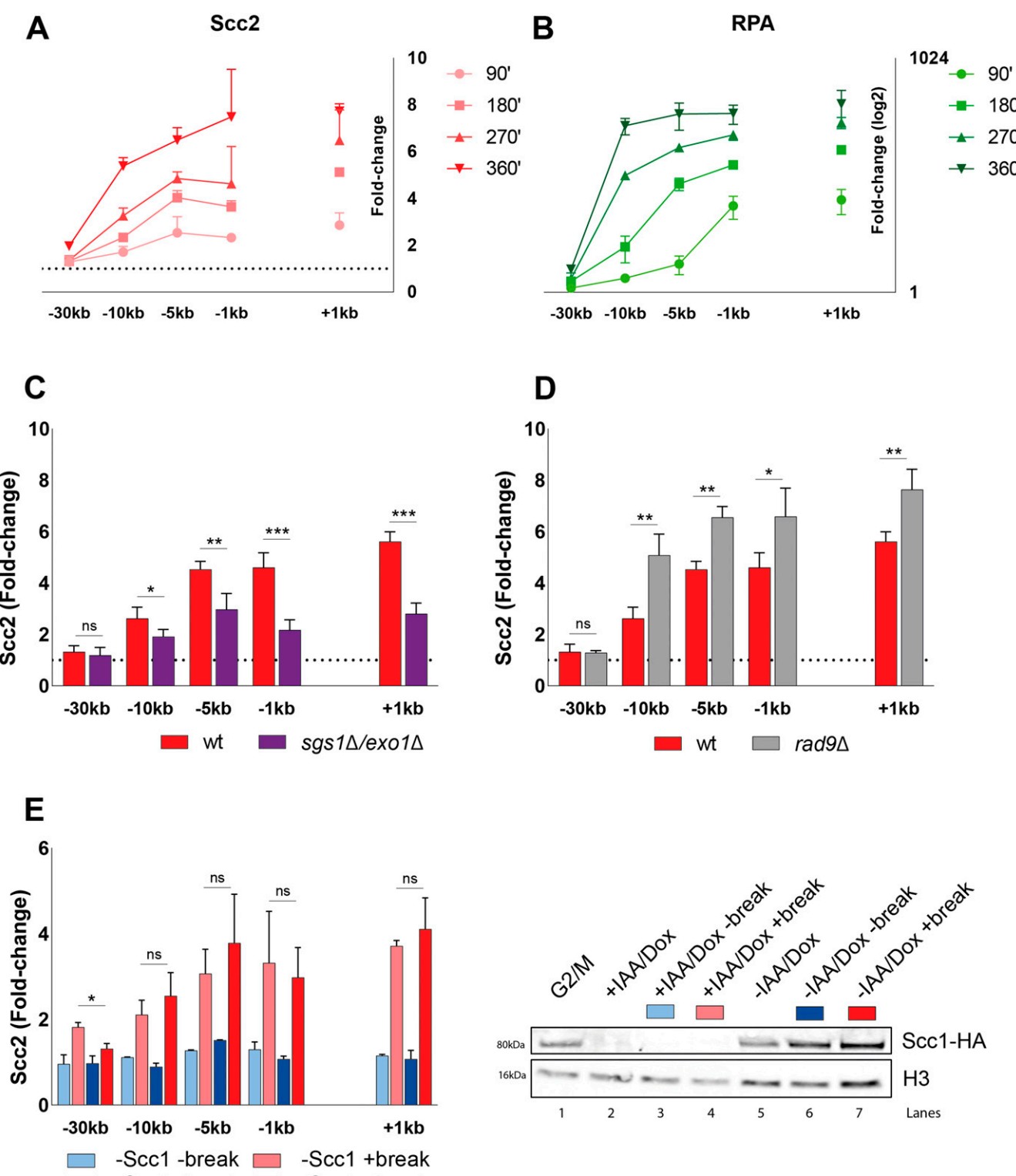

**Figure 4. Scc2 binding at double-strand breaks accumulates over time and depends on DNA end resection.**
**(A, B)** Chromatin immunoprecipitation (ChIP)-qPCR time course of (A) Scc2 and (B) RPA binding at the DSB (93% at 90' and >99% onward). Samples were taken over a 6-h period at 90 min intervals. A $\log_2$ scale visualizes RPA binding. **(C, D)** ChIP-qPCR of Scc2 binding at a DSB in (C) wild type and $sgs1\Delta exo1\Delta$ (>99% and 86%); (D) wild type and $rad9\Delta$ (both >99%). **(E)** Left: ChIP-qPCR of Scc2 binding at a DSB in the presence or absence of Scc1 in an Scc1 degron strain (>99% and 96%). Cells were grown and arrested as indicated in Fig 1. Before break induction, cultures were split in two, with one half receiving auxin and doxycycline and the other half the corresponding amount of EtOH for 2 h, to degrade Scc1 or not. Each culture was then split again, totaling four and receiving galactose or not. Right: Western blot showing protein levels of Scc1. Protein samples were taken after 3 h arrest (G2/M, lane 1), subsequent 2 h of either IAA/Doxy (+IAA/Dox, lane 2), or EtOH (–IAA/Dox, lane 5) incubation and following 3 h of

previous experiments, RPA itself does not facilitate Scc2's recruitment. A consequence of the reduced affinity of RPA for ssDNA in the *rfa1-G78E* background could result in diminished Mec1 recruitment. Although we have not verified this experimentally, this possibility would support our previous observation that Mec1 is dispensable for Scc2 recruitment. Furthermore, the modest effect of *rad51Δ* on Scc2 suggests that its recruitment is reliant on events preceding Rad51 filament formation.

Overall, our data suggest that the recruitment of Scc2/4 to DNA DSBs occurs during, and is dependent on DNA end resection. It further relies on phosphorylation of histone H2A by Tel1, but not on Mec1. Although affected by deficient chromatin remodeling, we believe that this does not extend beyond the impact chromatin remodeling has on DNA end resection, as the reduction in Scc2 recruitment was comparable with the reduction in RPA coverage. Whereas Scc2/4's binding remains confined to the proximity of the DSB and follows DNA end resection, cohesin is loaded at these sites but translocates to more distal regions and, contrary to Scc2/4, the loading of cohesin does require functional Mec1.

## Discussion

Cohesin's accumulation at DNA DSBs and its general dependency on Scc2/4 for its chromatin loading are both well documented. However, research in the field of DNA damage repair has been focusing almost exclusively on cohesin (69). To get mechanistic insight into how cohesin is loaded at DSBs it is therefore indispensable to understand how its loader gets there in the first place. Here we provide the first investigation focusing on the recruitment of Scc2 to DNA DSBs in budding yeast. We find that its accumulation depends mainly on γH2A and DNA end resection, neither of which alone suffices to facilitate recruitment. Although cohesin and its loader share several factors needed for their accumulation at DSBs, our study also uncovered an unexpected difference between the recruitment of Scc2 and the loading of cohesin. Whereas both Tel1 and Mec1 are required for de novo loading of cohesin at DSBs, Mec1 is dispensable for the recruitment of Scc2.

The significance of DNA end resection for HR based repair is well established, yet only recently its impact on cohesin is starting to gain traction (49, 70). We show that Scc2 recruitment emanates from DSBs coincident with ongoing resection. Similar observations have been made for chromatin remodelers modulating DNA end resection (48); however, we did not find evidence for Scc2's involvement in this process (Fig 3C). Supported by the fact that Scc2's binding at DSBs was significantly reduced in a *sgs1Δexo1Δ* mutant, this points towards a unidirectional relationship between the recruitment of Scc2 and DNA end resection.

In accord with previous data for cohesin, we find that the MRX complex is required to facilitate the recruitment of Scc2 to DSBs. This dependency most likely relies on MRX' ability to recruit Tel1 and the resection machinery (71), as recruitment of Scc2 was comparable with the wild type in the nuclease-deficient *mre11-D56A and mre11-H125A* mutants which still allow complex formation (72, 73) (Fig S2A). It was shown that "clean" DSBs, meaning breaks without DNA adducts, can bypass the need for the initial incision at DNA ends by MRX and Sae2 to promote Dna2 and Exo1 (74), which would also explain why deletion of Sae2 had no effect on the accumulation of Scc2 at DSBs (Fig S2B).

Most strikingly, we find that deletion of Mec1 has no effect on Scc2 recruitment, yet impairs cohesin loading at the DSB. The exact nature of cohesin's dependency on Mec1 is still unknown. It was suggested that phosphorylation of Scc1 at Ser83 by Chk1, presumably downstream of Mec1, was required for the generation of damage induced cohesion, yet loading around the break was unaffected by an S83A mutation (43). Likewise, Scc3 was found to be phosphorylated by Mec1, both in response to DNA damage as well as an unperturbed cell cycle (75).

Although both Scc2 and Scc4 harbor multiple consensus motifs for Mec1/Tel1 (76), we were unable to detect phosphorylation of these sites in response to DNA damage by mass spectrometry (data not shown), dampening the possibility of a direct effect on Scc2/4 by either. In vitro experiments have demonstrated that second strand capture of cohesin is favored if the target is single-stranded. These events were counteracted by addition of RPA (66). Applying this concept on a DSB, it can be envisioned that Mec1 phosphorylates RPA (77), destabilizing its association with DNA (78) and thereby enabling the loading of cohesin by Scc2/4. This is indeed supported by our finding that Scc2 accumulation at the DSB is increased in cells harboring the *rfa1-G77E* mutant compared with the wild type. It could also be that recruitment of Mec1 affects the chromatin accessibility around the break, as has been observed at stalled replication forks (79), which in turn favors the loading of cohesin (54).

The requirement of γH2A for the recruitment of Scc2 is consistent with what has been observed for cohesin (20). However, previous studies have shown that the *hta1-S129A* background causes accelerated end resection (80), indicating that DNA end resection by itself is insufficient for Scc2 recruitment. Conversely, it was also shown that γH2A spreading increases in the absence of Sgs1/Exo1 (81), indicating that also γH2A alone is insufficient for Scc2 recruitment. The fact that recruitment of Scc2 was increased in a *rad9Δ* background, likewise shown to have accelerated resection kinetics but an unaltered γH2A profile (5), lead us to the conclusion that recruitment is not directly mediated by DNA end resection but rather augmented by it. However, although end resection determines Scc2 recruitment, it cannot solely account for cohesin's accumulation around the break as cohesin levels increase well

---

either break induction in the presence (−IAA/Dox +break, lane 7) or absence of Scc1 (+IAA/Dox +break, lane 4) or under no break condition in the presence (−IAA/Dox −break, lane 6) or absence of Scc1 (+IAA/Dox −break, lane 3). Histone H3 served as a loading control. **(A, B, C, D, E)** Graphs show means and SD of (A, B, E) n = 2 and (C, D) n = 3. **(C, D)** *t* test was used to compare values of Scc2 between wild type and (C) *sgs1Δexo1Δ* or (D) *rad9Δ* at respective locations, 180 min after break induction. **(E)** In (E) binding of Scc2 in +break was compared between the presence or absence of Scc1 at respective locations, 180 min after break induction. Significance: *P < 0.05; **P < 0.01; ***P < 0.001; ns, not significant. Data were adjusted to the average cut efficiency for respective strain shown in squared brackets. Source data are available online for this figure.

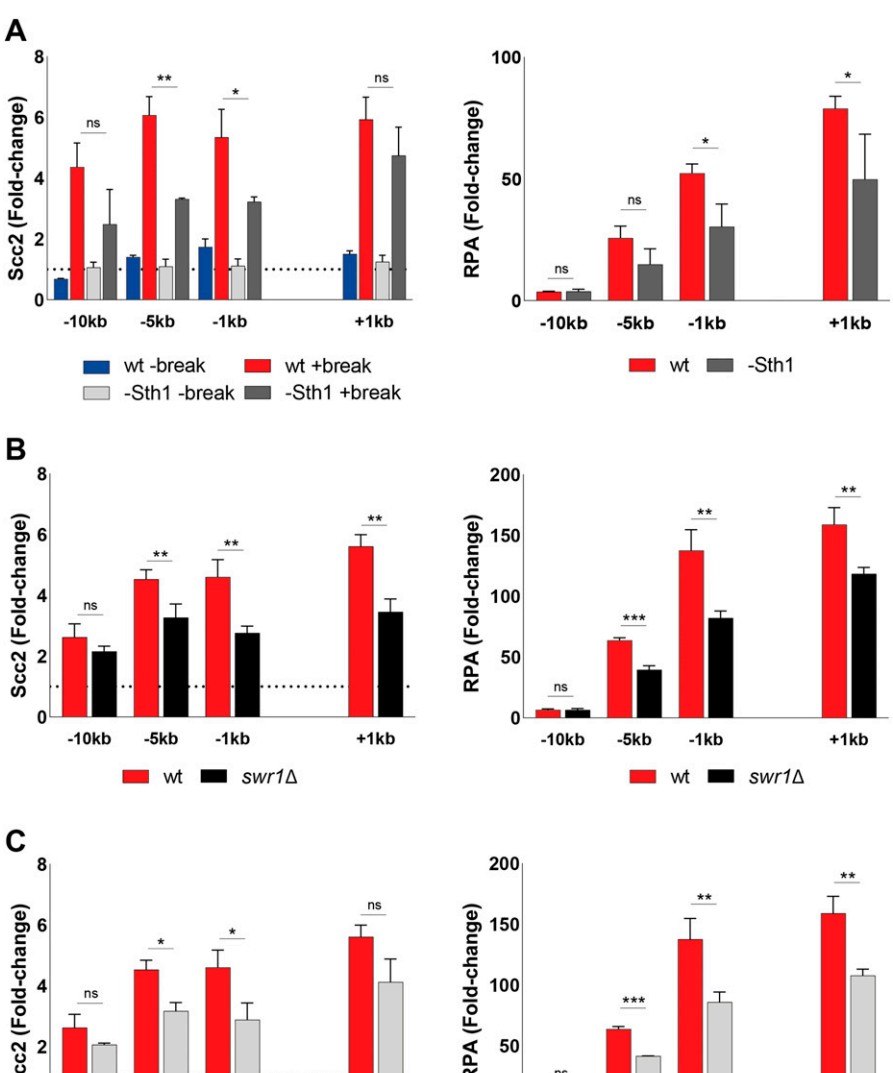

**Figure 5. Scc2 recruitment is not directly facilitated by RSC, SWR1, or INO80.**
**(A)** Left: Chromatin immunoprecipitation (ChIP)-qPCR of Scc2 binding at the double strand break (DSB) site in a wild type (97%) and a Sth1-AID (85%) strains in the presence or absence of DSB induction. All cells were grown in –met media and shifted to benomyl containing YEP media supplemented with 2 mM Methionine for G2/M arrest during 3 h. Cultures were then split, and received galactose or not. Right: ChIP-qPCR of RPA in the wild type and Sth1-AID strains at the DSB. Cells were grown as for the left graph. **(B)** ChIP-qPCR of Scc2 (left) or RPA (right) binding at a DSB in wild type and *swr1Δ* (>99% and 89%). **(C)** ChIP-qPCR of Scc2 (left) or RPA (right) binding at a DSB in wt and *nhp10Δ* (>99% and 91%). **(A, B, C)** The graphs show means and SD of (A) n = 2, (B, C) n = 3. **(A, B, C)** *t* test was used to compare normalized values of Scc2 between (A) wild type and –Sth1 in the presence of a break (B, C) wild type and indicated mutants at respective locations, 180 min after break induction. In addition, a *t* test was used to compare normalized values of RPA between wild type and indicated mutant. Significance: *P < 0.05; **P < 0.01; ***P < 0.001; ns, not significant. Data were adjusted to the average cut efficiency for respective strain shown in squared brackets.

beyond resected DNA. Our observation that a transition-state SMC3 mutant accumulates within 1 kb of the break could shed some light on this phenomenon. As ATP hydrolysis was shown to be required for cohesin's translocation along DNA (14), it can be envisioned that cohesin is loaded by Scc2/4, which is recruited during the process of end resection, and then translocates away from its loading site. As cohesin and γH2A show largely overlapping binding profiles in response to a DSB (20), γH2A could in this case be interpreted as a signpost for cohesin's movement. A study conducted on stalled replication forks found that cohesin ubiquitylation by the Rsp5 ubiquitin ligase enables mobilization of cohesin (82). Interestingly, this process was driven by Mec1. It could therefore be envisioned that Mec1 does not enable cohesin loading per se but allows its relocation from the site of loading either through modification of cohesin and/or phosphorylation of histone H2A. However, this notion warrants deeper investigation beyond the aim and scope of

this study. We also cannot exclude the possibility of interspersed loading sites between our investigated loci.

Because of the complex interplay of DNA end resection and chromatin remodeling, we reasoned that chromatin remodelers could dictate the recruitment of Scc2 depending on the biological context, as previous studies have demonstrated for Scc2 under unchallenged conditions (54). Given the role of the RSC complex in the DNA damage response (55) and the requirement of RSC components for cohesin loading at DSBs (83), we expected similar results for Scc2. Although recruitment in the absence of Sth1 was reduced overall compared with genuine wild type cells (Fig 5A), there was still a considerable increase in Scc2 loading in response to the DSB, arguing against Sth1 serving as an Scc2/4 loading factor also at DSBs. We reason that this reduction is rather due to impaired DNA end resection, as demonstrated by hindered RPA binding (Fig 5B).

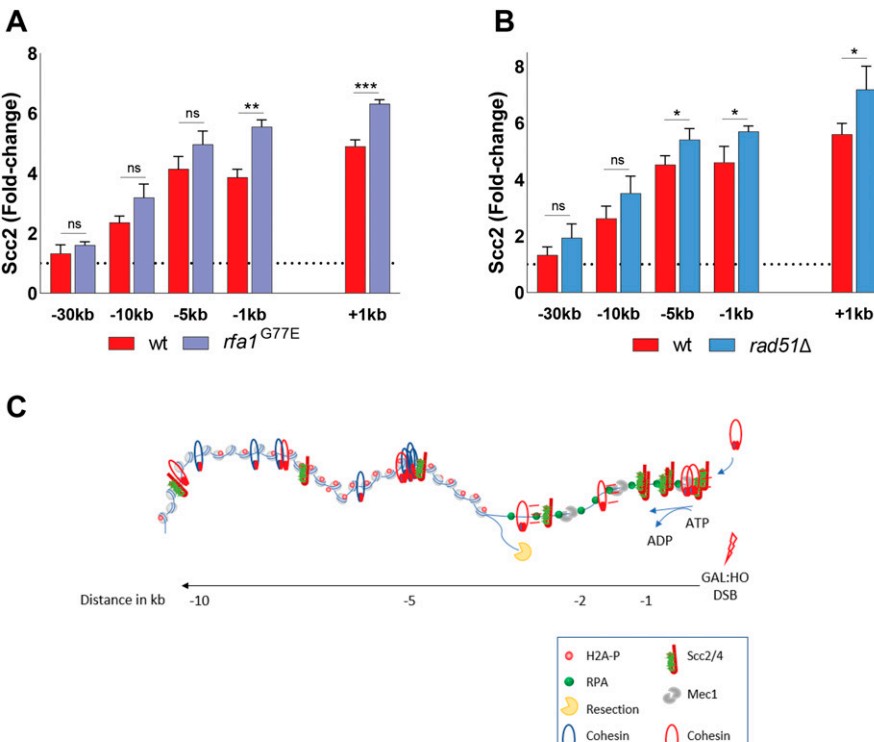

**Figure 6. Scc2 competes with RPA and Rad51 for binding to resected double strand break ends.**
**(A, B)** Chromatin immunoprecipitation-qPCR of Scc2 binding at the DSB, 90 min after break induction in a wild type (>99%) and (A) an *rfa1*-G77E mutant strains (>99%), (B) a *rad51Δ* strain (>99%). **(A, B)** The graphs show means and SD of (A, B) n = 3. **(A, B)** *t* test was used to compare normalized values of Scc2 between (A, B) wild type and indicated mutants at respective locations, 180 min after break induction. Significance: *P < 0.05; **P < 0.01; ***P < 0.001; ns, not significant. Data were adjusted to the average cut efficiency for respective strain shown in squared brackets in respective figure legend. **(C)** A schematic summary illustrating the stage of DSB repair via HR where the MRX complex has initiated end resection and Tel1 has mediated H2A phosphorylation. The resulting recruitment of Scc2/4 then facilitates cohesin loading at DSB ends provided Mec1 is present. Subsequently, ATP hydrolysis allows cohesin translocation to more distal sites.

Based on our finding that Scc2 recruitment depends on γH2A, we decided to focus on SWR1-C and INO80-C, both of which have been proposed to depend on γH2A (57, 59), although this claim has been contested (81). SWR1-C was demonstrated to be recruited to breaks facilitating the incorporation of H2A.Z (60), whereas INO80-C catalyzes its removal in addition to general nucleosomal eviction (84). Although in vitro experiments have demonstrated that incorporation of H2A.Z benefits DNA end resection, the absence of Swr1 has reportedly no resection defect in vivo (48, 85). These findings are not reflected by our results as we do see delayed RPA binding in our *swr1Δ* strain (Fig 5B, right). However, this discrepancy could depend on timing for break induction because ongoing break induction in a similar experimental setup raised the possibility of a minor resection defect (59). An alternative reason could be slower HO kinetics (61). Because our data demonstrate that the levels of Scc2 at the DSB correlate with time (Fig 4A), delayed break induction would make Scc2, and RPA, binding "lag" behind. Although the level of break induction in our *swr1Δ* strain was comparable with the wild type after 3 h (Fig S4C), we cannot exclude the possibility that the break kinetics differ from wild type in our experimental conditions.

The absence of Nhp10 resulted in a reduction of Scc2 recruitment on par with the results obtained for Sth1. Confirming previous reports, DNA end resection was hampered in this strain (Fig 5C, right), affecting Scc2 recruitment in a likewise manner (Fig 5C, left), supporting the hypothesis of DNA end resection being a decisive factor. Based on these data, we reached a similar conclusion as with Sth1 and believe that neither SWR1-C nor INO80-C are directly responsible for recruitment of Scc2 as its binding correlated well with RPA coverage. However, their impact on DNA end resection

does affect Scc2's presence at the DSB. Nevertheless, as our investigation did not comprehensively address all chromatin remodelers, we cannot exclude the possibility that other complexes are responsible for Scc2/4 recruitment at DSBs.

The exact mechanism that facilitates the recruitment of Scc2 to DSBs remains to be determined. Although Scc2/4 was shown to be bind poorly to ssDNA in vitro, its affinity for Y-fork DNA was comparable with dsDNA (28). It can be envisioned that in the process of end resection, a similar intermediate is formed, favoring its recruitment. We have previously demonstrated that inactivation of Scc2 in yeast modulates transcription globally and in response to a DSB, affecting DSB proximal genes in particular (26). Studies in human cells have shown that transcriptional repression at DSBs is mediated by NIPBL and cohesin (86), whereas in yeast, this process has been credited to DNA end resection (80). Accumulating evidence highlights the significance of RNA and transcription in the DNA damage response and the modulation of resection (87). Considering our data, it would be interesting to address the impact of transcription at DSBs on Scc2 recruitment and vice versa (88).

In summary, we demonstrate that recruitment of Scc2 relies on phosphorylation of H2A by Tel1 and the subsequent resection of DNA. Based on this we propose that DNA end resection affects the loading of cohesin at DSBs in two ways. First, the actual resection process mediates the recruitment of Scc2/4. Second, the subsequent recruitment of Mec1 enables Scc2/4 to load cohesin at DSB ends, whereupon ATP hydrolysis allows cohesin translocation to more distal sites (Fig 6C). Together, these data provide a more detailed insight into the events which facilitate the recruitment of Scc2 and subsequent accumulation of cohesin at DNA DSBs.

# Materials and Methods

## Yeast strains and growth conditions

All *S. cerevisiae* strains were of W303 origin (ade2-1 trp1-1 can1-100 leu2-3 his3-11,15 ura3-1 RAD5, GAL, and psi+). Yeast extract peptone (YEP) supplemented with 40 µg/ml adenine was used as yeast media, unless otherwise stated. For chromatin immunoprecipitation experiments, cells were grown in YEP media supplemented 2% raffinose at 25°C, unless otherwise stated. Arrest in G2/M was induced by addition of benomyl (381586; Sigma-Aldrich) dissolved in DMSO at a final concentration of 8 µg/ml, for 3 h, and break induction achieved by the addition of 2% galactose (final), or not during indicated time periods. Where applicable, 3-indoleacetic acid (auxin—I3750; Sigma-Aldrich) was dissolved in 100% EtOH and added at a final concentration of 1 mM. Doxycycline (D9891; Sigma-Aldrich) was dissolved in 50% EtOH and added at a final concentration of 5 µg/ml. Control samples received the respective amount of EtOH. To create null mutants, the gene of interest was replaced with an antibiotic resistance marker through lithium acetate based transformation. Some strains were crossed to obtain desired genotypes. For a complete list of strains, see Table S1.

## FACS analysis of DNA content

G2/M arrest was confirmed by flow cytometric analysis. In brief, 1 ml of cultures were fixed overnight in 70% EtOH. Samples were resuspended in 50 mM Tris–HCl, pH 7.8, and treated with RNAse A (12091021; Thermo Fisher Scientific) shaking at 37°C overnight. Cells were resuspended in FACS buffer (200 mM Tris, pH 7.5, 211 mM NaCl, and 78 mM MgCl$_2$) containing propidium iodide (P4170; Sigma-Aldrich) and sonicated using a Bioruptor Standard (UCD-200; Diagenode). Samples were analyzed on a BD FACSCalibur (BD Biosciences) using the CellQuest Pro software.

## Protein extraction and Western blotting

To verify auxin-mediated degradation of target proteins, 4 OD units of cells were collected, washed with water, and resuspended in glass bead disruption buffer (20 mM Tris–HCl, pH 8.0, 10 mM MgCl$_2$, 1 mM EDTA, 5% glycerol, and 0.3 M ammonium sulfate) supplemented with 1 mM DTT, cOmplete protease inhibitor (Roche), and 1 mM PMSF. 0.8 g of acid washed glass beads (G4649; Sigma-Aldrich) were added and samples vortexed on a VXR Basic Vibrax (Thermo Fisher Scientific) for lysis. Samples were run on Bolt 4–12% Bis-Tris Plus gels (NW04120BOX; Thermo Fisher Scientific) before transfer to nitrocellulose membranes (GE10600002; Sigma-Aldrich). Anti-FLAG (F1804; Sigma-Aldrich), anti-cdc11 (y-415; Santa Cruz Biotechnology), anti-H3 (ab1791; Abcam), anti-AID tag (CAC-APC004AM-T; 2B Scientific), and anti-HA (ab137838; Abcam) antibodies were used in conjunction with appropriate secondary antibodies from the IRDyes series (LI-COR) and detected on an Odyssey imaging system (LI-COR).

## Chromatin immunoprecipitation quantitative PCR (ChIP-qPCR)

ChIP was performed as described (89), with some modifications. 40 OD units of cells were crosslinked in 1% formaldehyde for 30 min at room temperature, washed twice in 1× cold TBS, frozen in liquid nitrogen after resuspension in lysis buffer (50 mM Hepes-KOH, pH 7.5, 140 mM NaCl, 1 mM EDTA, 0.1% Na-deoxycholate, 1% Triton X-100, 1× cOmplete protease inhibitor [Roche], and 1 mM PMSF), and mechanically lysed using a 6870 freezer/mill (SPEX, CertiPrep). Whole Cell Extracts were sonicated using a Bandelin Sonopuls HD 2070.2 mounted with an MS73 probe, for optimally sized DNA fragments (300–700 bp). The protein of interest was purified by overnight incubation with anti-FLAG (F1804; Sigma-Aldrich) or anti-RFA antibody (AS07 214; Agrisera), coupled to Dynabeads protein A (Invitrogen). Samples were then washed successively 2× with lysis buffer, 2× with lysis buffer (360 mM NaCl), 2× wash buffer (10 mM Tris–HCl, pH 8, 250 mM LiCl, 1 mM EDTA, 0.5% Na-deoxycholate, and 0.5% NP-40) and once with TE buffer. After elution of samples from the beads in elution buffer (50 mM Tris–HCl, pH 8, 10 mM EDTA, and 1% SDS) at 65°C for 15 min, crosslinking was reversed for both IP and input samples at 65°C overnight. After 1 h RNAse (VWR) and 2 h Proteinase K (Sigma-Aldrich) treatment, DNA was purified using a QIAquick PCR Purification Kit (QIAGEN). Analysis of DNA was performed by qPCR using Fast SYBR Green Master Mix (Applied Biosystems) on a 7500 FAST Real Time PCR System (Life Technologies). Where applicable, data were normalized to an average of N1 and N2 within the same sample. For a list of primers, see Table S2.

## Measurement of ssDNA at resected DNA ends

10 ml of cells (OD = 0.7) were collected and resuspended in 500 µl of extraction buffer (100 mM NaCl, 50 mM Tris–HCl, pH 8.0, 10 mM EDTA, and 1% SDS) supplemented with 2 µl β-mercaptoethanol (M6250; Sigma-Aldrich) and 2.5 µl of Zymolase 100T (20 mg/ml). Cells were lysed for 30 min at 37°C followed by 5 min at 65°C. 250 µl KOAc was added followed by incubation on ice for 20 min. The lysate was centrifuged and the supernatant was mixed with 0.2 ml of 5 M NH$_4$OAc and 1 ml isopropanol. The resulting pellet was dissolved in 100 µl of TE and 200 µl isopropanol, washed with 80% EtOH, and resuspended in 50 µl of TE. 10 µl of each sample were digested with 10 U of AciI (R0551S; New England Biolabs) and 10 U MseI (R0525S; New England Biolabs) in a total reaction volume of 30 µl using CutSmart Buffer (27204S; New England Biolabs) supplemented with 1 µl of Ambion RNAse A (AM2271). The digestion was performed overnight at 37°C. Undigested control samples were treated equally with the omission of restriction enzymes. Concentration was measured and adjusted if necessary. Five serial dilutions of 1:5 were made for each sample and then quantified using Fast SYBR Green Master Mix (Applied Biosystems) on a 7500 FAST Real-Time PCR System (Life Technologies). For a list of primers see Table S2. The difference in average cycles (ΔCt) between digested and undigested samples was measured and the amount of ssDNA calculated according to (47).

$$\%ssDNA = 100/\left(\left(1 + 2^{\Delta Ct}\right)/2\right).$$

## Pulsed-field gel electrophoresis (PFGE) and Southern blot

Break induction at the HO cut site was confirmed with PFGE. The procedure was carried out as previously described (19). Briefly, cells were collected and fixed overnight in 70% EtOH at −20°C. Samples

were resuspended in resuspension buffer (1 M Tris base, pH 7.5, 1.2 M sorbitol, and 0.5 M EDTA) and lysed in SEMZ buffer (1 M Sorbitol, 50 mM EDTA, 28 mM β-Mercaptoethanol, and 2 mg/ml Zymolyase 100T [IC320932; VWR]), at 37°C for 90 min. Plugs were then prepared with SEZ buffer (1 M Sorbitol, 50 mM EDTA, and 1 mg/ml Zymolyase 100T) and 1% low-melting temperature agarose (A9414; Sigma-Aldrich). Embedded cells were then lysed in EST buffer (10 mM Tris, pH 8, 100 mM EDTA, and 1% sarcosyl) at 37°C for 45 min. After successive equilibration in 0.5× TBE, plugs were loaded on a 1% PFGE agarose (1620137; Bio-Rad) gel prepared in 0.5× TBE. Chromosomes were separated on a Biorad Chef DIII (Bio-Rad) at 6 V/cm with a 35.4–83.6 s switch time and 120° included angle for 24 h. Gels were subsequently subjected to Southern blot using standard techniques. The PCR product of primers "−1 kb Chr VI cut Fw" and "−0.3 kb Chr VI cut Rv" served as probe for the break site. A loading control probe for chromosome V was generated using primers "Southern Chr V Ctrl Fw" and "Southern Chr V Ctrl Rv" (Table S2). Cut efficiency was determined by densiometric analysis of Cut and Uncut Chr VI bands in relation to the Chr V loading control band.

## Supplementary Information

## Acknowledgements

We thank Professors C Björkegren, F Uhlmann, J Downs, K Nasmyth, and L Symington for strains and plasmids. This work was supported by the Swedish Research Council (2016-02206), the Swedish Cancer Society (2016/554, 2019/410), and the Bergvall Foundation (2016-01868, 2017-02287) to L Ström, and the KID program at the Karolinska Institutet to L Ström for M Scherzer (2-3591/2014).

### Author Contributions

M Scherzer: conceptualization, data curation, formal analysis, validation, investigation, visualization, methodology, project administration, and writing—original draft, review, and editing.
F Giordano: conceptualization, formal analysis, investigation, and methodology.
MS Ferran: formal analysis, investigation, and methodology.
L Ström: conceptualization, resources, data curation, supervision, funding acquisition, investigation, visualization, methodology, project administration, and writing—review and editing.

### Conflict of Interest Statement

The authors declare that they have no conflict of interest.

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
