## [Reviewer comments · Life Science Alliance]

Life Science Alliance

Recruitment of Scc2/4 to double strand breaks depends on γ H2A and DNA end resection

Martin Scherzer, Fosco Giordano, Maria Ferran, and Lena Ström

DOI: <https://doi.org/10.26508/lsa.202101244>

Corresponding author(s): *Lena Ström, Karolinska Institute*

Review Timeline:

Submission Date:	2021-09-24
Editorial Decision:	2021-09-24
Revision Received:	2021-12-10
Editorial Decision:	2022-01-07
Revision Received:	2022-01-13
Accepted:	2022-01-14

Scientific Editor: *Eric Sawey, PhD*

Transaction Report:

Please note that the manuscript was reviewed at *Review Commons* and these reports were taken into account in the decision-making process at *Life Science Alliance*.

Review
COMMONS

September 24, 2021

Re: Life Science Alliance manuscript #LSA-2021-01244-T

Dr. Lena Ström
Karolinska Institutet
Cell and Molecular Biology
Berzelius väg 35
Kvarter 7B
Stockholm 171 77
Sweden

Dear Dr. Ström,

Thank you for submitting your manuscript entitled "Recruitment of Scc2/4 to double strand breaks depends on γ H2A and DNA end resection" to Life Science Alliance. We invite you to re-submit the manuscript, revised according to your Revision Plan.

Thank you for this interesting contribution to Life Science Alliance. We are looking forward to receiving your revised manuscript.

Sincerely,

Eric Sawey, PhD
Executive Editor
Life Science Alliance
<http://www.lsa-journal.org>

B. MANUSCRIPT ORGANIZATION AND FORMATTING:

Manuscript number: #LSA-2021-01244-T

Corresponding author(s): Lena Ström

Response to comments from reviewers on:

“Recruitment of Scc2/4 to double strand breaks depends on γ H2A and DNA end resection”, by Martin Scherzer et al

We would like to thank the editors and reviewers for their time spent, as well as their appreciated and insightful comments on our manuscript. We here outline point by point how we have revised the manuscript in response to the remarks from reviewers.

Reviewer #1 (Evidence, reproducibility and clarity (Required)):

In the manuscript entitled "Recruitment of Scc2/4 to double strand breaks depends on γ H2A and DNA end resection", Scherzer et al. study the role of Scc2 in DSB repair in yeast. Scc2 is part of the cohesin loader and it is required for cohesin loading in response to DSB. The authors study the chromatin association of Scc2 by ChIP-qPCR and use genetics to identify factors that affect its recruitment. They show that Scc2 is enriched up to 10 kb from the break site, similar to cohesin and identify MRE, TEL1 and γ H2A as important factors for Scc2 chromatin binding. Remarkably, MEC1 that has been shown to regulate cohesin under these conditions is dispensable for Scc2 recruitment. While DNA resection is important for Scc2 recruitment, chromatin remodelers don't play a significant role in it despite numerous reports on their effect on cohesin loading during the cell cycle. The manuscript provides new and important information on cohesin regulation in response to DNA damage.

****Major comments:****

The experiments are done appropriately and contain the required control. The results are presented clearly and with adequate statistics and support the conclusions. The experiments provide valuable information. However, the low resolution of the experimental setup is limiting, and dynamic information of Scc2 binding is lacking. I would agree with the authors that this kind of information may be beyond their scope. However, the absence of this information reduces the overall impact of the manuscript.

1. ChIP-seq, of at least some of the key experiments, could provide information on the specific Scc2 binding sites and elucidate whether cohesin is translocated from the loading sites or accumulate in its proximity.

ChIP-seq would indeed increase the resolution of the Scc2 and Cohesin DSB accumulation, especially beyond 1 kb. However, to gain insight into the dynamics of the binding, numerous timepoints for both strains would have to be analyzed, which we feel would be beyond the possibilities for this study. For Scc2 we believe that we have shown high enough resolution, determining binding from 0,1 to 30 kb away from the break. We have also provided a time course experiment from 90 minutes up to 6 hours and show that the Scc2 binding is

*continuously increasing, at all sites analysed, see Fig 4A. We have in the revised version of the manuscript added experiments looking at the Cohesin binding in close vicinity of the break – similar to what we previously did for Scc2. We have also compared Cohesin binding at 90 and 180 min after break induction, for increased information on the dynamics of its binding at the DSB, and see no change in Cohesin positioning in relation to the DSB site. Rather the general level of binding increases equally over the region, with time (compare Fig 1B and 4A with Fig 1C and Fig S3). This to us indicated that either there is no translocation of Cohesin from the site(s) of loading to the final binding sites. Alternatively, translocation is happening relatively fast and not possible to capture when performing ChIP on wt Cohesin. To further clarify this, we have now included ChIP qPCR experiments on HA-tagged SMC3 and the ATPase hydrolysis deficient *smc3E1155Q*, previously found to be proficient in DNA binding but prevented from translocation (Hu et al 2010, “ATP Hydrolysis is required for relocating Cohesin from sites occupied by its Scc2/4 loading complex”). These experiments indicate that Cohesin is indeed loaded at DSB ends and then translocated to its final binding sites with time. These data have been added to the manuscript in Supplementary Fig S1C and S1D.*

2. It has been suggested that Scc2 and Pds5 are mutually exclusive in cohesin complexes. It would be interesting to check in the current experimental setup (ChIP-qPCR) if Pds5 is mimicking Scc2 pattern

We generated a strain where Pds5 was FLAG-tagged, and now include experiments determining the loading/binding of Pds5 in the break region. These show (Fig S1B) that binding of Pds5 mimics that of Cohesin, indicating that it binds as part of the Cohesin complex. In addition, it is seemingly not affected by the presence of a DSB.

****Minor comments:****

1. Adding a threshold line to the graphs at fold change= 1 (no enrichment in respect to wild type) will increase their readability.

We appreciate this suggestion, this has now been added, and is indeed helpful.

2. Fig. 1A- Add times to the schematic. Modify the text to GAL addition/break induction.

Thank you for the good suggestion, the figure has now been modified.

3. Page 9. The authors write: "Cohesin failed to be loaded at the DSB in a *mec1Δ* background (Fig 3A)". However, the figure shows reduced cohesin binding in *mec1delta* in respect to the wild type.

*In this graph Cohesin binding in response to break induction is shown. The level of binding in the *mec1* deletion mutant is comparable to that of Cohesin in the absence of break induction, See Fig S3 for a newly added experiment showing wt binding of Cohesin at the same timepoint. The text describing Fig 3A on page 10 has now also been slightly modified.*

4. Page 10. ".....recruitment to the DSB compared to wild type (Fig 3D)."Should be Fig. 4D.

Thank you for noticing this mistake, this has now been corrected.

5. Figure legend 3. "Protein samples were taken after 3 hours arrest (G2/M, lane 1),....." The

benomyl arrest is referred to as G2 arrest in the text but G2/M arrest in the legend. Consistency is needed.

We agree on the importance of consistency and have thus changed to G2/M throughout the manuscript.

I suggest presenting the suggested model in a figure

We have added a hopefully comprehensive and illustrative model figure as Fig 6C in the revised version of the manuscript.

Reviewer #1 (Significance (Required)):

I am an expert in cohesin biology.

The Scc2-Scc4 complex has been identified as an essential factor for cohesin loading during the cell cycle (Ciosk et al., 2000). This function has been shown to be essential for cohesin role in response to DNA DSB (Unal et al., 2004, Strom et al., 2004). The interplay between Scc2 and the cohesin has been studied mostly in the context of the cell cycle. It has been shown that Scc2 activates the ATPase activity of cohesin and promotes its translocation from the loading site. Scc2 and Pds5 are mutually exclusive and their switch suppresses cohesin ATPase activity (Hu et al., 2011, Petela et al., 2011). However, the Scc2-cohesin interplay has been poorly studied in the context of DNA repair. The current work adds valuable information on the factors that recruits Scc2 to the break site and identifies end resection as the key event in this process. This information is novel and important and its contribution to the fields of cohesin and DNA repair should not be overlooked. However, ChIP-seq information can increase the overall impact.

We appreciate the nice verdict. We do agree to some extent on the ChIP seq comment, however based on the rationale described in our response to major point 1, we have instead provided results from experiments utilizing an ATP-ase deficient Cohesin mutant.

Reviewer #2 (Evidence, reproducibility and clarity (Required)):

Cohesin is a key structural component of chromosomes. Amongst its functions, cohesin plays a critical role in ensuring the accurate repair of double stranded DNA breaks (DSBs). Intuitive as this may seem, a number of fundamental open questions remain. One of these questions is, how does the cohesin loading machinery recognise a DSB? This issue is addressed in the present study. The manuscript begins with a well-written introduction into the fields of DSB repair, as well as cohesin. The research aim is clearly laid out. Experiments follow that sequentially investigate known steps of the DSB repair pathway, asking how these steps intersect with the cohesin loading machinery.

On the positive side, this is a technically very well conducted study (investigating the cohesin loader has proven tricky in many contexts). The study is systematic and explores the known steps during DSB repair for their impact on cohesin loader recruitment. The authors find a surprising separation

of function. The DSB pathway up until H2AX phosphorylation and DNA end resection is required for both cohesin loader recruitment, as well as consequently for cohesin loading. The Mec1 checkpoint kinase, in contrast, is dispensable for cohesin loader recruitment but is required for cohesin loading. This suggests that Mec1 supports cohesin loading at a step beyond that of attracting the cohesin loader. The manuscript thus contains important information that will be of interest to a wide range of researchers in the DNA repair and cohesin fields.

The limitation of the study lies in the fact that the molecular determinant for cohesin loader recruitment to DSBs remains unknown. H2AX phosphorylation and DNA end resection are shown to be prerequisites, but how do these events form a molecular mark that the cohesin loader recognises? And what is this mark? Equally, how does the Mec1 kinase permit cohesin loading additionally to the cohesin loader?

We appreciate the positive comments as well as the criticism, and agree on the lack of precise knowledge regarding the actual mark made by phosphorylation of H2A, and resection, for recruitment of Scc2. The same is true for the limited understanding of what the exact contribution of Mec1 for Cohesin loading is. Despite this we believe that we provide a large number of valuable experiments, revealing new information about the similarities and differences in requirements for recruitment of Cohesin and its loader at a DSB.

****Specific comments:****

Figure 1. It would be interesting to overlay the Scc2 profile around the DSB next with that of Scc1 (obtained previously under similar conditions?), to contrast the loading site with the final cohesin distribution.

In the revised version of the manuscript, we have looked at the binding of Cohesin close to the break and outwards in the same way as for Scc2, with our experimental system. These binding profiles are not overlapping, shown as Fig 1B and 1C as well as Fig 4A and Fig S3. Their different distribution is very clear. This also confirms what has been reported previously for Cohesin binding, where the region closest to the break is in principle devoid of Cohesin (Fig 1C). This binding pattern is also not changed with increased time for break induction (Fig S3) when studying wt Cohesin. This to us indicated that either there is no translocation of Cohesin from the site(s) of loading to the final binding sites. Alternatively, translocation is happening relatively fast and not possible to capture when performing ChIP on wt Cohesin. To further clarify this, we have now included ChIP-qPCR experiments on HA-tagged SMC3, and the ATPase hydrolysis deficient smc3E1155Q, previously found to be proficient in DNA binding but prevented from translocation (Hu et al 2010, "ATP Hydrolysis is required for relocating Cohesin from sites occupied by its Scc2/4 loading complex"). These experiments indeed indicate that Cohesin is actually loaded at DSB ends and then translocated to its final binding sites with time. These data have been added to the manuscript in Supplementary Fig S1C and S1D.

Figure 2. Using the same y-axis scale from 1-4 amongst panels A-D could make evaluation of the data easier.

We agree the comparison is made easier when the scale is the same - this has now been changed within figures.

Figure 3. Panels A and B contain data that are important to interpret the DNA end resection results shown in Figure S2. Maybe that latter data, which conveys the main conclusion from the figure, could be incorporated within the main figure?

This is a good point and we have changed accordingly, now resection experiments in the absence of Scc2 from Fig S2 are shown as Fig 3C.

Figure 5. In this figure, the authors begin to investigate possible contributions of candidate cohesin loader receptors, in the form of chromatin remodelling complexes. The Swr1 and INO80 remodellers have an effect on DNA end resection that parallels the effect on Scc2 recruitment, suggesting that their main contribution might be that of facilitating DNA end resection.

This relationship remains less well documented in the case of Sth1 depletion. Both when using the sth1-3 allele, or degron depletion, the authors observe a relative reduction of cohesin loader recruitment, compared to what they would otherwise expect. However, in both cases a side-by-side analysis of a similarly-treated wild type strain is missing. Whether or not RSC inactivation impacts cohesin loader recruitment therefore remains uncertain.

In the revised version of the paper we have included experiments where wild-type cells were grown in the same culturing system as the Sth1 degron strain, included as Figure 5A. The best control would of course be to use the Sth1 degron strain and not degrade Sth1 as the wild-type control. However, the poor growth of these cells in -Met media with raffinose as the sole carbon source is not compatible with the experimental design.

For the experiment on the ts allele of Sth1, we agree that a comparison with a wild-type control would be desirable. However, the wild-type control was not possible to keep arrested in G2/M at restrictive temperature during the course of the experiment. We therefore decided to omit the data from the ts strain in the manuscript.

It is also not documented what the corresponding effect of RSC inactivation on DNA end resection might be. Given that previous results suggested that RSC might contribute to cohesin loading at DSBs, the nature of how RSC does this could maybe be clarified before publication.

In the revised version of the manuscript we are including RPA ChIP data for the Sth1 – degron strain (Figure 5B). These show that resection is moderatley, albeit significantly, reduced after degradation of Sth1. We believe this to be the explanation for the reduced Scc2 loading in its absence, in line with what is seen in the swr1 and nhp10 deletion mutants.

Reviewer #2 (Significance (Required)):
see above.

Reviewer #3 (Evidence, reproducibility and clarity (Required)):

This paper presents data analysing the recruitment of Scc2 to double strand breaks. It makes the interesting observation that its recruitment is Tel1 but not Mec1 dependent, and does not require remodelers (it seems). It does correlate with resection but the mechanism of loading is unclear.

I have a few issues on controls and alignment of text with results in this manuscript. Also there is some omission of important recent work and some old studies. But if these points can be resolved it could be published.

****Major points:****

1. The cut efficiency under all conditions tested needs to be presented and the CHIP needs to be normalized in every assay to the cut efficiency. This is particularly relevant in the mutants of remodelers as they definitely influence the efficiency of Gal-HO induction. This must be included for every chip result.

We agree that the Cut efficiency could influence the degree of recruitment, not the least due to the strength of the signal from the break for recruitment of the initial DSB response factors that we show are important for recruitment of Scc2. Already in the previous version we therefore showed (Figure S3C) that the cut efficiency of the chromatin remodelers was comparable to that in WT cells after 3 hours (new figure S4B). We have now either repeated this type of experiment three times for strains used in the study and calculated an average cut efficiency for each strain or used an RT-PCR based method for quantification of the Cut efficiency on the actual ChIP samples, when available. The obtained results have then been used for normalization of the ChIP-qPCR results. The average Cut efficiency is indicated for each strain in the respective figure legends in the new version of the manuscript. Normalization of the ChIP data to the Cut efficiency does not change the principle results or conclusions presented previously, throughout the manuscript.

2. The arp8 delta mutant is clearly polyploid and probably has some suppressor mutation or another problem. They should discard the arp8 results and get a proper and controlled arp8 delta strain (from another lab in europe - there are several with good W303 strains).

When preparing the initial arp8Δ strains, we transformed several experimental strains, which all resulted in different degree of diploidy on a population basis. Loss of INO80 components has previously been shown to confer diploidy or polyploidy in an S288C background, excluding Arp8 (Chambers et al., Genes Dev. 2012). Considering the apparent differences regarding INO80 (the INO80 ATPase subunit is essential in W303 but not in S288C), we deemed it plausible that di/polyploidization could be a resulting phenotype of an Arp8 deletion in W303.

Prompted by the comments put forward here we however transformed a clean W303 wild-type strain and indeed saw no sign of di/polyploidy. However, combining arp8Δ with GAL:HO, in the presence or absence of an extra recognition sequence for HO by crossing, consistently lead to diploidy. While searching for suitable arp8Δ mutants, we noticed that adequate strains (van Attikum et al., EMBO J 2007 & Cell 2009, Lademann et al, Cell Rep 2017) originated from JKM179. The difference between JKM179 and our W303 strains is, to the best of our knowledge, the absence of the HML/HMR loci in the former. We hypothesize that this is the reason for our diploidization phenotype. INO80's and in particular Arp8's general impact on transcription has been well studied (Poli et al., Philos Trans R Soc Lond B Biol Sci. 2017), and we believe that deletion of Arp8 causes a marginal leakage in the GAL promoter. As our strains harbor both the HML/HMR loci and an intact MAT locus, this occasionally leads to mating type switch, which in a population results in subsequent mating and diploid zygotes, not possible in

JKM179. While this could potentially be interesting to investigate further, it lies in our view outside the scope of this study. Experiments in the hmlΔ /hmrΔ genetic background would not be in line with all other experiments included in our study, and we therefore decided to not employ this. We agree that the diploidization does not hold up to experimental standards and in this version of the manuscript the arp8Δ experiments have therefore been removed. We want to extend our thanks to the reviewer to point this out.

3. The text does not accurately reflect the results in several places. For instance .. on page 10 where the result of sgs1 exo1 mutant strain is described, it is said that "Recruitment of Scc2 to the DSB was drastically reduced.... and "consistent with long range resection the effect was less prominent closer to the break.". First, the word "drastic" is not appropriate for a drop of about 50% (on average) and in reality the drop is more significant near the cut (+1kb) than far from the break (+ 10 or 30 kb).... - the data are the opposite of what is stated. and it is not drastic. I do not contest that it correlates with resection, if the HO-cut efficiency is equal in all strains.

We are sorry for this discrepancy between the results shown and the description of the same in a few cases. We have rephrased the results section to reflect the data more accurately. We have also removed the sgs2exo1 deletion mutant data close to the break as we have not investigated all mutants in the region closest to the break and thereby lack a comprehensive comparison.

4. The results with INO80 and SWR1 are not really compelling - what is the cut efficiency in these strains. Moreover, the "confusion" in the literature is only because people look at different loci and different conditions. INO80 does affect resection (see Van Attikum et al., 2007; and Cheblal A et al., Molecular Cell 2020) for resection assays in wt and mutant strains. And it is very strange that the Van attikum et al., Cell 2004 (the back to back paper with Morrison et al Cell 2004) is not cited. The data on resection is clear in this early work. But it appears that the arp8 mutant used has other mutations and polyploidization, and should clearly be discarded. Nhp10 impact is a bit controversial but not arp8 with a good strain.

The references in general are missing Cheblal A et al., Molecular Cell 2020 for Cohesin recruitment, impact on resection and arp8 impact and ditto. Also missing is Deshpande I et al., molecular Cell 2017 for RPA-Ddc2-Mec1 interactions. These omissions are strange and in fact create confusion in the ms.

We would like to thank the reviewer for bringing our attention on some very relevant articles published in the field that has now been referenced as we hope correctly. We have in the revised version of the manuscript also adjusted the ChIP-qPCR results to the average efficiency of break induction, which is indicated in respective Figure legend.

****Minor points:****

The english usage needs to be corrected at a few places... and figures are not correctly cited always - see page 10 especially - there is no Figure 3D.

We have gone through the text carefully, and also asked a native English speaker to assess the language, and corrected accordingly. We are sorry for the Figure mistake, this has now been corrected together with a general update of figure numbers based on some modifications of the manuscript structure.

Reviewer #3 (Significance (Required)):

The advance is not groundbreaking but still interesting and worthy of publishing, if proper controls and better referencing can be done.

We hope that we after having related all ChIP-qPCR data to averaged Cut efficiencies for each strain, and edited the discussion to relate it more appropriately to both new and older correct references, have been able to handle the issues raised.

January 7, 2022

RE: Life Science Alliance Manuscript #LSA-2021-01244-TR

Dr. Lena Ström
Karolinska Institute
Cell and Molecular Biology
Berzelius väg 35
Kvarter 7B
Stockholm 171 77
Sweden

Dear Dr. Ström,

Thank you for submitting your revised manuscript entitled "Recruitment of Scc2/4 to double strand breaks depends on γ H2A and DNA end resection". We would be happy to publish your paper in Life Science Alliance pending final revisions necessary to meet our formatting guidelines. Please address the remaining minor comments from Reviewers 2 and 3.

- please add the Twitter handle of your host institute/organization as well as your own or/and one of the authors in our system
- please consult our manuscript preparation guidelines <https://www.life-science-alliance.org/manuscript-prep> and make sure your manuscript sections are in the correct order
- please add an Author Contributions section to your main manuscript text
- please add a conflict of interest statement to your main manuscript text
- please use capital letters when introducing the panels in their legends
- there is a callout for figure S3A in the manuscript text. Please revise
- please indicate molecular weight next to each protein blot
- if the vertical line in Figure S4A indicates a cut in the gel, please mention this in the figure legend

A. FINAL FILES:

B. MANUSCRIPT ORGANIZATION AND FORMATTING:

Sincerely,

Reviewer #1 (Comments to the Authors (Required)):

The revised paper is improved and is now acceptable in my opinion for publication in LSA.

Reviewer #2 (Comments to the Authors (Required)):

I have now read the revised manuscript. The authors have made an exemplary effort in addressing all my concerns with additional informative experiments. These have greatly improved this manuscript that I can now highly recommend for publication.

There is one final point that I'd like to give the authors for consideration. This regards the interpretation of the experiment to inactivate the RSC chromatin remodeller. The authors include new data to confirm that DNA end resection is affected following RSC inactivation, which is a likely reason for reduced Scc2 recruitment. Whether or not RSC also plays a direct role in Scc2 recruitment, like at undamaged chromosomal loading sites, has not been addressed by this experiment (this would require restoring end resection in the absence of RSC). I therefore suggest a slight rephrasing of the authors' conclusion from this experiment, especially in the abstract:

"Although affected by their impact on resection, the recruitment of Scc2 appears to not be directly facilitated by chromatin remodeling complexes." A statement that excludes a direct role for RSC is not supported by the data. Rather a statement that suggests a positive role of end resection, without excluding an additional direct role, would be a more accurate conclusion.

Reviewer #3 (Comments to the Authors (Required)):

In the revised manuscript "Recruitment of Scc2/4 to double strand breaks depends on γ H2A and DNA end resection", the authors adequately addressed my concerns. The revised manuscript is interesting and solid. It provides new information about cohesin function in DNA repair and will be valuable to the cohesin community and should be published.

I suggest a few minor edits:

I accept the authors' argument that dynamics experiments are beyond the scope of this work and the current data provides, although not in high resolution, sufficient data to support their conclusions. Therefore, no other experiments are needed. The newly added data, describing PDS5, SMC3 and smc3-E1155Q, showed in sup fig 1C-D is important and interesting. Therefore, I suggest moving them from the sup figure to the main text. It shows, surprisingly, no increase in DSB dependent PDS5 binding. The result suggests that PDS5 might be excluded from the complex under these condition. The SMC3 ChIP reveals another interesting point. The SMC3 and SCC1 DNA binding profiles are different, and the Smc3 profile represents a merge of the Scc2 and Scc1 profiles. The minor issues were fully addressed by the authors. I suggest adding to the model in fig 5C a scale showing the distance from the DSB.

January 14, 2022

RE: Life Science Alliance Manuscript #LSA-2021-01244-TRR

Dr. Lena Ström
Karolinska Institute
Cell and Molecular Biology
Solnavägen 9
Biomedicum, Kvarter 7B
Stockholm 171 77
Sweden

Dear Dr. Ström,

Thank you for submitting your Research Article entitled "Recruitment of Scc2/4 to double strand breaks depends on γ H2A and DNA end resection". It is a pleasure to let you know that your manuscript is now accepted for publication in Life Science Alliance. Congratulations on this interesting work.

DISTRIBUTION OF MATERIALS:

Again, congratulations on a very nice paper. I hope you found the review process to be constructive and are pleased with how the manuscript was handled editorially. We look forward to future exciting submissions from your lab.

Sincerely,
